# Nondestructive Testing and Visualization of Catechin Content in Black Tea Fermentation Using Hyperspectral Imaging

**DOI:** 10.3390/s21238051

**Published:** 2021-12-02

**Authors:** Chunwang Dong, Chongshan Yang, Zhongyuan Liu, Rentian Zhang, Peng Yan, Ting An, Yan Zhao, Yang Li

**Affiliations:** 1Tea Research Institute, The Chinese Academy of Agricultural Sciences, Hangzhou 310008, China; dongchunwang@163.com (C.D.); yang1029345485@163.com (C.Y.); lzy13112350601@163.com (Z.L.); z8t13040542337@163.com (R.Z.); yanpengzn@163.com (P.Y.); anting_mac@163.com (T.A.); 2College of Mechanical and Electrical Engineering, Shihezi University, Shihezi 832000, China

**Keywords:** congou, fermentation, catechin component content, hyperspectral, quantitative forecast, visual analysis

## Abstract

Catechin is a major reactive substance involved in black tea fermentation. It has a determinant effect on the final quality and taste of made teas. In this study, we applied hyperspectral technology with the chemometrics method and used different pretreatment and variable filtering algorithms to reduce noise interference. After reduction of the spectral data dimensions by principal component analysis (PCA), an optimal prediction model for catechin content was constructed, followed by visual analysis of catechin content when fermenting leaves for different periods of time. The results showed that zero mean normalization (Z-score), multiplicative scatter correction (MSC), and standard normal variate (SNV) can effectively improve model accuracy; while the shuffled frog leaping algorithm (SFLA), the variable combination population analysis genetic algorithm (VCPA-GA), and variable combination population analysis iteratively retaining informative variables (VCPA-IRIV) can significantly reduce spectral data and enhance the calculation speed of the model. We found that nonlinear models performed better than linear ones. The prediction accuracy for the total amount of catechins and for epicatechin gallate (ECG) of the extreme learning machine (ELM), based on optimal variables, reached 0.989 and 0.994, respectively, and the prediction accuracy for EGC, C, EC, and EGCG of the content support vector regression (SVR) models reached 0.972, 0.993, 0.990, and 0.994, respectively. The optimal model offers accurate prediction, and visual analysis can determine the distribution of the catechin content when fermenting leaves for different fermentation periods. The findings provide significant reference material for intelligent digital assessment of black tea during processing.

## 1. Introduction

Black tea is known to be the most consumed tea in the world. It is made by complete fermentation and contains large amounts of phenols; it offers a unique taste, and antioxidant and lipid-lowering effects. Fermentation is a critical process for generating the splendid color, aroma, and taste of black tea. The cell tissues of fresh tea leaves are destroyed after rolling treatment, and enzymatic action of polyphenols takes place under specific temperature, humidity, and oxygen-content conditions. The cytoplasmic inclusions of the leaves undergo oxidation, polymerization, and condensation reactions, forming colored substances and special aromatic substances [1]. Catechin is a phenolic active substance extracted from natural plant material such as tea leaves, and is a major contributor to tea astringency. Catechin constitutes 70–80% of the total polyphenols in tea, and contains five catechin monomers (EGC, C, EC, EGCG, and ECG) which can be divided into ester and non-ester catechins. Ester catechins account for 80% of the total catechins. They exhibit stronger astringency and bitterness, while non-ester catechins have weaker astringency but a refreshing aftertaste [2]. Therefore, changes in catechin content play a determinant role in the color and taste of brewed tea as well as the grade of black tea.

During fermentation, the inclusions gradually decompose and transform through processes such as enzymatic action, oxidation and reduction of polyphenols, hydrothermal action, and acidic action. The chemical reactions are complicated. The most obvious results on the macro level are the changes to the aroma and color. The aroma gradually changes from a grassy smell to a cooked smell, and the leaf color turns from green to a purplish bronze. The catechin content can be measured by HPLC (high performance liquid chromatography) to assess the fermentation status [3]. However, this method is costly and requires a lengthy testing period, and therefore does not allow real-time quantitative prediction. Currently, black tea manufacturers or workshops normally use traditional methods to determine the fermentation status of black tea. This method of examining the color and aroma of black tea using the sensory judgment of tea masters is very subjective, making it hard to ensure the quality of different batches of tea; it would not be compatible with intelligent digital processing of black tea [4].

Hyperspectral imaging technology can fuse image data with spectral data, detect geometric spatial distribution information on fermentation, and achieve complete characterization of target items. Hyperspectral imaging is becoming a core approach in the assessment of agricultural products [5]. However, the amount of hyperspectral information acquired is huge, and processing is complicated. Therefore, it is very important to determine a convenient and stable modeling method to quickly and accurately detect sample information. Yu-Jie Wang et al. [6] applied a successive projections algorithm (SPA) to extract the characteristic wavelengths of the hyperspectral data collected for fresh tea leaves, and established a successive projections algorithm with multiple linear regression (SPA-MLR) model by combining the SPA with an MLR (multinomial logistic regression) algorithm. The model can be used for rapid detection of the moisture, total nitrogen, and crude fiber content in fresh tea leaves, and to determine a quality index score. The prediction accuracy of the model reached 0.9357, 0.8543, 0.8188, and 0.9168, respectively. Wei et al. [7] collected hyperspectral images of both the front and back of tea leaves on a conveyor belt in the laboratory to simulate the actual production environment. They designed a logistic regression classifier to identify the spectra of the front and back of fresh tea leaves, then imported the adjusted spectra into the least square support vector regression (LS-SVR) model to predict the water content of the tea leaves. Results showed that the accuracy for predicting the front and back water content reached 0.9500 and 0.9560, respectively, and the RMSEPs (root-mean-square errors of prediction) were 0.028 and 0.027, respectively. Lin Yuan et al. [8] regarded 542, 686, and 754 nm as the sensitive bands for anthracnose in tea plants, based on the hyperspectral data of tea leaves, and they implemented a strategy of unsupervised classification and an adaptive two-dimensional threshold for disease recognition. The overall recognition rate reached 98%. Dong Chunwang et al. [9] applied electrical characteristics detection technology to detect the catechin content of black tea during fermentation. They compared the abilities of different pretreatment and variable filtering algorithms to eliminate noise. The prediction accuracy of variable combination population analysis—iteratively retains informative variables—random forests algorithm (VCPA-IRIV-RF) model in detecting catechin content reached 0.988.

The above studies show that hyperspectral technology has been widely applied in tea processing and tea plant characteristics detection. However, no studies on the detection of the catechin content of black tea during fermentation have been reported. Therefore, this study was designed with the following objectives: (1) completion of a correspondence analysis between hyperspectral data of black tea with different fermentation periods and physicochemical test values of catechin content; (2) discussion of the relationship between the position of fermenting leaves and the catechin content for different fermentation periods; (3) comparison of the denoising effect of different pretreatment and variable screening algorithms and establishment of an optimal model for the prediction of catechin levels; (4) visualization of the distribution of catechin content in black tea at different fermentation levels.

## 2. Materials and Methods

### 2.1. Acquisition of Experimental Samples

The variety of the fresh tea leaves used in this study was Tieguanyin. The tenderness was one bud with one leaf, and we used a batch of tea leaves weighing 15 kg. The samples were picked at the Tea Research Institute of the Chinese Academy of Agricultural Sciences, Shengzhou Base. The experiments were performed at the processing building of the Tea Research Institute. The technique used was the traditional method for making Congou black tea: fresh tea leaves → withering → rolling → fermentation → drying [10]. The fermentation device was a manual climatic box. The fermentation temperature was set at 30 °C, relative humidity was ≥90%, and the fermentation time was 5 h, to make sure the samples underwent all the fermentation levels (mild fermentation, moderate fermentation, and excessive fermentation) [11]. The acquisition of hyperspectral data on the samples was performed concurrently with the sample fixing [12]. An ultra-low-temperature freezer was used as the fixing machine. In order to investigate the changes in catechin content of stacked fermenting leaves at different positions and to improve the generalization performance of the model, we performed layer-based processing on the stacked leaves. The fermented leaves were divided into three layers, each with a length and width of 0.5 m × 0.3 m and a thickness of 5 cm. The total number of tea samples was 18.

### 2.2. Detection of Catechin Content

Grinding pretreatment was performed on the freeze-dried samples. Standard catechin components were purchased from Sigma-Aldrich (St. Louis, MO, USA). The specific content was determined as per the standard “*Determination of Total Polyphenols and catechins content in Tea*” (*GB*/*T8313-2008*).

### 2.3. Specific Operations for Component Detection

The reaction liquid consisted of 70% methanol, 2% ethanoic acid mobile phase (phase A), and acetonitrile mobile phase (phase B). To prepare the 2% ethanoic acid mobile phase (phase A), we used purified water with 40 mL of ethanoic acid to make a constant volume of 2 L of liquid, shook it, filtered it with a sand core funnel, and applied ultrasonic debubbling for 15 min. To prepare the acetonitrile mobile phase (phase B), we simply applied ultrasonic debubbling for 15 min to 500 mL of purified acetonitrile. To prepare the mother liquor, we ground the sample evenly, put a 0.2 g sample into a 10 mL centrifuge tube, mixed it with 5 mL of 70% methanol solution, and placed it in a 70 °C water bath. After soaking it for 10 min, we cooled it to room temperature, centrifuged it for 10 min at a speed of 3500 r/min, then saved the centrifuged supernatant in a volumetric flask. The residue was extracted repeatedly according to the above procedures to make a 10 mL extraction solution, and then filtered with a 0.45 um membrane for standby application [13].

A Shimadzu LC-20AD HPLC (Shimadzu Co., Kyoto, Japan) equipped with a 20AV UV–Vis (ultraviolet–visible) detector was used. The chromatographic column settings were: C18-BDS column (Hangzhou Coulomb Technology Co., Ltd., Hangzhou, China) (4.6 mm × 200 mm); sample size: 10 uL; column temperature: 35 °C; column temperature limit: 37 °C; flow rate: 10 mL/min; gradient elution: mobile phase A was reduced from 93.5% to 85% at 0–16 min and 85% to 75% at 16–25 min, then increased from 75% to 93.5% at 25–25.5 min, and became balanced for 5 min at 25.5–30 min; mobile phase B was increased from 6.5% to 15% at 0–16 min and 15% to 25% at 16–25 min, then reduced from 25% to 6.5% at 25–25.5 min, and became balanced for 5 min at 25.5–30 min [14].

The standard sample required calibration after every 10 samples were tested, and the catechin content required three repeated tests. The specific content was calculated with the following formula:catechin content (%) = (A ∗ V ∗ C)/(A1 ∗ m ∗ 10^3^ ∗ m1) ∗ 100(1)
where A is the peak area of the test-liquid sample, V is the volume of the test liquid (10 mL), C is the catechin concentration in the sample (mg/mL), A1 is the peak area of a standard sample, m is the dry-matter content in the sample, and m1 is the mass of the sample.

### 2.4. Acquisition of Hyperspectral Data

The hyperspectral data acquisition system consisted of a hyperspectral imager (ImSpector V10E, Spectral Imaging Oy Ltd., Oulu, Finland), a high-precision conveyor platform, and a set of 150 W optical fiber halogen lamps which were fixed in a camera obscura with a 45-degree distribution. Before data acquisition, the machine was preheated for 30 min; we then set the resolution at 2.8 nm, the sampling interval at 0.67 nm, the exposure time at 4.2 ms, the sample conveyor speed at 1.24 mm/s, the spacing between the sample and the lens at 22.6 cm, and the light intensity at 103 cd. After acquisition, we carried out black and white plate correction again, following the methods of Yang Chongshan et al. [15]. The process of data acquisition is shown in Figure 1.

During data acquisition, the samples were evenly placed in a 17 cm × 8 cm rectangular glass frame with the surface of the tea leaves flush with the top of the glass frame. Tea leaves at fermentation time 0 were used as the initial sample. The sampling interval was 1 h, with one sampling for each layer (upper, middle, and lower). The spectral data acquisition was performed using ENVI 5.3 (64-bit), and the specific process was as follows: a rectangular region formed by 200 × 200 pixels was used as the region of interest (ROI), and at each moment, hyperspectral data were collected for the upper, middle, and lower layers. We selected 10 symmetrically distributed ROIs for each hyperspectral image. The average of the spectra in each ROI were used as sample spectra for that moment, and each moment contained 30 spectra, for a total of 180 spectra [16].

### 2.5. Data Treatment and Analysis

#### 2.5.1. Data Standardization

During data acquisition, machine noise, an uneven sample surface, and the external environment may lead to deviations in the spectral data. Therefore, in order to reduce interference and improve the prediction accuracy of the model, we pretreated the original spectra using seven algorithms: Smooth, SNV, Multiplicative scatter correction (MSC), 2 Derivative (2 De), Center, Center and Zero-mean normalization (Z-Score), and Min-Max normalization (Min-Max).

#### 2.5.2. Variable Screening and Dimension Reduction

Various bands of spectral data were apparent after acquisition. Before establishing a specific component prediction model, it was necessary to use different algorithms to select the spectral bands closely related to specific components, in order to eliminate surplus data and improve the prediction accuracy and calculation speed of the model. Therefore, we used the seven above-mentioned variable screening algorithms to extract the characteristic wavelengths. After variable screening, PCA was conducted to perform quadratic dimensional reduction compression on the selected data and to optimize the model’s accuracy [17]. The characteristic wavelengths screened for different endoplasmic components are shown in Table 1.

#### 2.5.3. Modeling Algorithms and Evaluation Criteria

The optimal pretreatment algorithm and optimal variables of the spectra were determined based on the sensitivity degree of the specific components, and a prediction model for catechin content was established using PLS, ELM, and SVR models. The input of the principal components of the models was determined using minimal root-mean-square error cross-validation (RMSECV). The evaluation indices included root-mean-square error of prediction (RMSEP), correlation coefficient of calibration set (Rc), correlation coefficient of predication set (Rp), and relative percentage deviation (RPD) [17]. Usually, the closer the RMSECV and RMSEP values, the better the generalization performance of a model. RPD is the ratio of standard deviation to RMSEP, which reflects the final performance of the model. When RPD is within a range of 1.4–1.8, the model can make a rough prediction for the sample; when RPD exceeds 2, the prediction of the model is accurate [18]. All data processing was completed in Origin 2018, ENVI 5.3, and MATLAB 2019a.

## 3. Results and Analyses

### 3.1. Change Trends of Catechin Content during Different Fermentation Periods

Figure 2 is a line chart of the physicochemical test results for catechin content in black tea during processing. As shown, polyphenols (mainly consisting of catechins) participated in the enzymatic oxidation and formed high polymers such as tea pigments. The reaction gradually changed from strong to moderate as fermentation continued. Thus, the quantity of catechins dropped dramatically over 0–2 h and then continued to decrease more slowly over 2–5 h [19]. Specifically, the EC component of the catechin content decreased by 96.28%, indicating that EC was the major component involved in the chemical reaction during fermentation. In contrast, the EGC content decreased by 4.5%, indicating that EGC was only slightly involved in the chemical reaction during the fermentation of black tea.

### 3.2. Comparison of Catechin Content at Different Periods and Different Positions of Fermenting Leaves

Figure 3 shows the changes in the catechin content in stacked, fermented black tea leaves at different fermentation levels (mild, moderate, and excessive) and positions (upper, middle, and lower). As the figure demonstrates, for the same fermentation position, the catechin content decreased as fermentation progressed, but the rate of decrease was different in different fermentation layers. This indicated that various chemical reactions were taking place during fermentation apart from enzymatic oxidation. At 2 h of fermentation, the catechin content was higher in the middle layer of fermenting leaves. The reason might be the low reaction rate of the middle layer, due to its lower oxygen content compared with the upper layer and its lower temperature compared with the lower layer. At 3–5 h, the catechin content was higher in the lower layer of fermenting leaves, as the enzymatic oxidation was greatly affected by the oxygen content.

### 3.3. Selection of Optimal Pretreatment Algorithms

Figure 4a shows the average hyperspectral curves for different tea fermentation periods. The endoplasmic components of black tea changed significantly during fermentation due to the effects of various chemical reactions. In the 700–957 nm band, the average spectra at different moments presented significant differences, indicating that different endoplasmic component contents could lead to different spectral absorbances. Hence, a prediction model can be built based on the spectral information. Figure 4b shows the original hyperspectral data, which include all the information on the black tea during fermentation as well as some of the sources of noise. Before modeling, the original data were denoised using different algorithms to improve the model accuracy. The pretreated spectral curves are shown in Figure 4c–h. We used a PLS model for the selection of different pretreatment methods, and the optimal method was determined by the Rp value of the models. The results are shown in Table 2.

### 3.4. Selection of Optimal Variable Screening Algorithms

We used a KS algorithm to divide the 162 samples into a calibration set and a prediction set, at a proportion of 3:1 [20]. Next, we imported the data sets into different screening algorithms to remove the surplus data, and determined the optimal variable screening method for catechin content by combining different methods with PLS model prediction. The results are shown in Table 3, revealing that the number of variables for the total catechins (EGC, C, EC, EGCG, and ECG) were reduced to 14, 32, 27, 24, 29, and 49, respectively. The surplus data elimination rate reached 91.20%, indicating a significant improvement in the prediction accuracy of the model.

The SPA algorithm was used to compare the vector magnitude based on vector projection analysis. The wavelength with the largest projection size was used as the candidate wavelength, and the selected combination of variables contained the minimum surplus information. Compared with the full-waveband PLS model, the SFLA-PLS model for total catechins was effective in increasing the Rc and Rp from 0.968 and 0.961 to 0.977 and 0.979, respectively. VCPA-GA adopted the binary matrix sampling method combined with a natural evolutionary process, and determined the number of wavelengths to be retained according to EDF to reduce the space of the variables [21]. Compared with the full-waveband PLS model, the VCPA-GA-PLS model for EGC content was able to increase the Rc and Rp from 0.810 and 0.769 to 0.946 and 0.956, respectively. The VCPA-GA-PLS model for ECG content increased the Rc and Rp from 0.989 and 0.983 to 0.996 and 0.994, respectively. Based on the concept of cluster classification, the VCPA-IRIV model divided the data sets into strong- and weak-information data, as well as interference information variables and uninformative variables. After multiple iterative computations, only strong- and weak-information variables were retained, to reduce the effect of interference information [22]. Compared with the full-waveband PLS model, the VCPA-IRIV model for the C content increased the Rc and Rp from 0.989 and 0.983 to 0.996 and 0.993; the VCPA-IRIV model for EC content increased the Rc and Rp from 0.983 and 0.971 to 0.996 and 0.990; and the VCPA-IRIV model for EGCG content increased the Rc and Rp from 0.988 and 0.980 to 0.996 and 0.994. Therefore, the quantitative prediction model for catechin content prediction was effective and can be used for content detection during actual tea production.

### 3.5. Selection of Optimal Models

After pretreatment of the obtained spectral data and elimination of noise, we employed different variable screening methods to select the characteristic wavelengths sensitive to specific components. Through PCA dimension reduction and with the number of principle components as the input, we established a linear PLS model and nonlinear ELM and SVR models [23]. ELM is a feedforward neural network learning method, which artificially gives a hidden layer node weight without the need for updating. This method is suitable for supervised and unsupervised learning to analyze the effect of different PCs on model performance [24]. The prediction performance is shown in Figure 5a,k. The SVR model constructed a hyperplane or a set of hyperplanes in high-dimensional or finite space for the data sets, thus realizing optimal segmentation by calculating the distance from the hyperplane to the training data point and making multiple-factor analysis easier. The SVR model uses an RBF radial basis function as the kernel function. Like any of the principal component input numbers, the penalty coefficient c and the kernel parameter g influence the accuracy of the model, so we used the verification function tunelsssvm for optimization. The results of optimization are shown in Figure 5c,e,g,i [25].

Figure 5a shows the prediction effect of the total catechin model, with the graph visualizing the relationship between the true value and the predicted value in the model prediction set. When PCs = 7, the model achieved the best prediction accuracy, and the Rp, RMSEP, and RPD values of the prediction set were 0.989, 0.175, and 6.5, respectively. The distribution scatter plot is shown in Figure 5b [26,27,28]. Figure 5c shows the optimization effect of the EGC model. The model had the best prediction accuracy when c = 2.83, g = 0.5, and PCs = 8. The Rp, RMSEP, and RPD values of the prediction set were 0.972, 0.0041, and 3.78, respectively. The distribution scatter plot is shown in Figure 5d. Figure 5e illustrates the optimization effect of the C model. The model had the best prediction accuracy when c = 16, g = 0.063, and PCs = 7. The Rp, RMSEP, and RPD values of the prediction set were 0.993, 0.0082, and 6.72, respectively. The distribution scatter plot is shown in Figure 5f. Figure 5g shows the optimization effect of the EC model. The model had the best prediction accuracy when c = 4, g = 0.35, and PCs = 7. The Rp, RMSEP, and RPD values of the prediction set were 0.990, 0.0075, and 5.69, respectively. The distribution scatter plot is shown in Figure 5h. Figure 5i displays the prediction effect of the EGCG model, which had the best prediction accuracy when c = 32, g = 0.63, and PCs = 5. The Rp, RMSEP, and RPD values of the prediction set were 0.994, 0.079, and 7.33, respectively. The distribution scatter plot is shown in Figure 5j. Figure 5k illustrates the optimization effect of the ECG model. This model had the best prediction accuracy when PCs = 7. The Rp, RMSEP, and RPD values of the prediction set were 0.994, 0.047, and 7.29, respectively. The distribution scatter plot is shown in Figure 5l.

### 3.6. Visualized Analysis of Catechin Content

The technological process of the fermentation experiment was controlled by professional tea masters. They divided the fermentation levels into three types according to the tenderness, withering, rolling, color, and aroma of the tea leaves. The first type was mild fermentation, which occurred at 0–3 h; the second was moderate fermentation, which occurred at 4 h; and the last was excessive fermentation, which occurred at 5 h [29]. The visual analysis was carried out using MATLAB software. During visual analysis, we selected the hyperspectral image for a 200 × 200 pixel region for each fermentation level, carried out binary processing on the wavelengths where the background information was obviously different from that of the ROI, and set the value of the background pixel points to zero. We then transformed the hyperspectral data into two-dimensional data, removed the previous and following noise wavelengths, denoised the primary variables, input the data to the ELM model for prediction, and finally, colored the data points to achieve visualization of the catechin content, the final visualization is shown in Figure 6. The change in the EGC content was not obvious during fermentation, so the visual analysis focused only on the distribution of the total catechin content (C, EC, EGCG, and ECG) at different fermentation levels.

## 4. Conclusions

(1) Physicochemical tests were performed on black tea samples with different fermentation periods, and the change trends of the various catechin contents were then analyzed. The results showed that all the contents except EGC exhibited a significant decrease (>72.06%) as enzymatic oxidation progressed during fermentation. The chemical reaction in black tea was complicated, and the catechin content at the same position but at different times showed different rates of decrease, due to other reactions. The catechin content of the middle fermenting layer had a slow conversion rate 2 h after fermentation due to insufficient oxygen and low temperature. The lower fermenting layer also had slow enzymatic oxidation because of lower oxygen content.

(2) Preliminary experiments proved that when the content of endoplasmic components in the black tea fermentation process was less than 0.01% and did not change significantly, it was impossible to accurately predict the endoplasmic components, even using a variety of modeling methods. The prediction accuracy of the proposed model improved significantly after denoising the original hyperspectral data using different pretreatment and variable screening methods. Nonlinear ELM and SVR models offered higher prediction accuracy than a linear PLS model. Due to multiple chemical reactions in fermenting black tea, as well as other factors, the catechin content shows inconsistent change trends; hence, nonlinear models are more accurate than linear ones for prediction.

(3) The prediction results of the models we tested showed that only the EGC model had a relatively low accuracy of 0.972. The optimal model prediction rates for total catechin contents (Total catechins, C, EC, EGCG, and ECG) were quite high at 0.989, 0.993, 0.990, 0.994, and 0.994, respectively, showing excellent prediction performance. After pretreatment, variable screening, and PCA dimension reduction on the original spectra, visual distribution of the catechin content of black tea at different fermentation levels was realized using the ELM model. We suggest that these results can provide a theoretical basis for intuitive and effective digital evaluation of the fermentation degree of black tea.

## Figures and Tables

**Figure 1 sensors-21-08051-f001:**
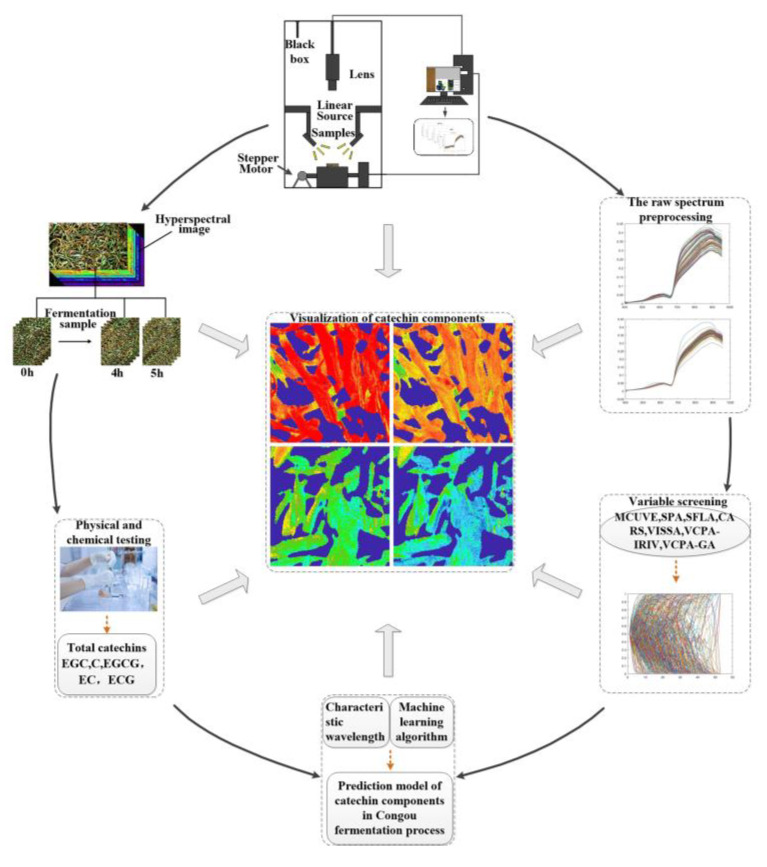
Hyperspectral data acquisition system flowchart.

**Figure 2 sensors-21-08051-f002:**
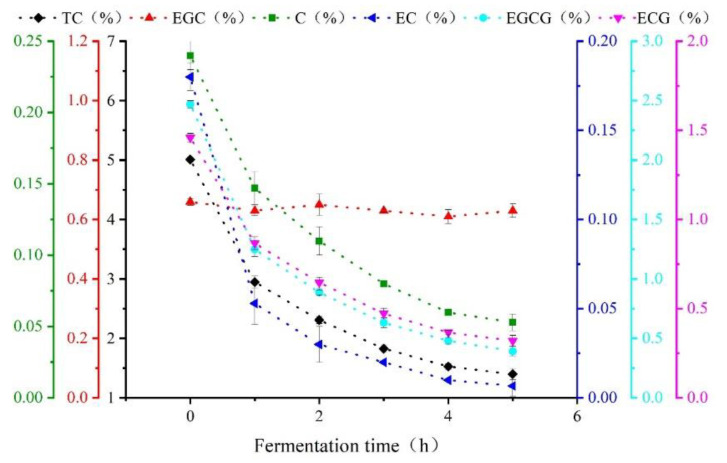
Change trends of catechin component content during Congou black tea fermentation.

**Figure 3 sensors-21-08051-f003:**
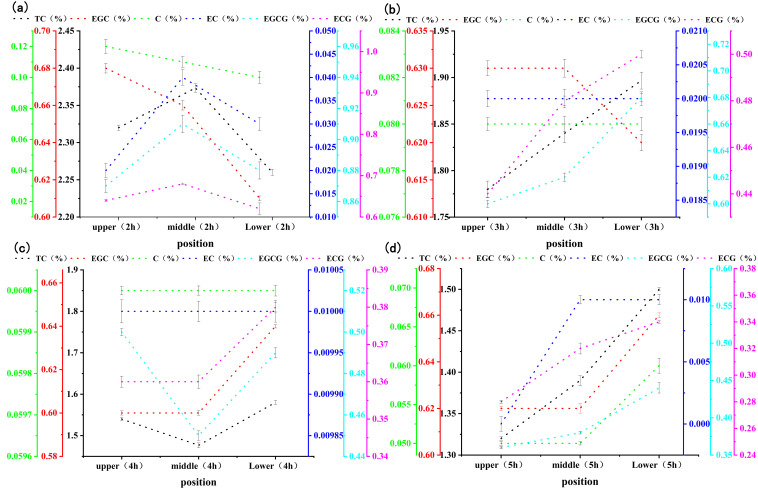
Changes in catechin content in the fermented leaves of stacked black tea at different fermentation levels and positions. ((**a**–**d**) are the changes in the content of catechin components at different positions during the fermentation of black tea for 2, 3, 4, and 5 hours, respectively).

**Figure 4 sensors-21-08051-f004:**
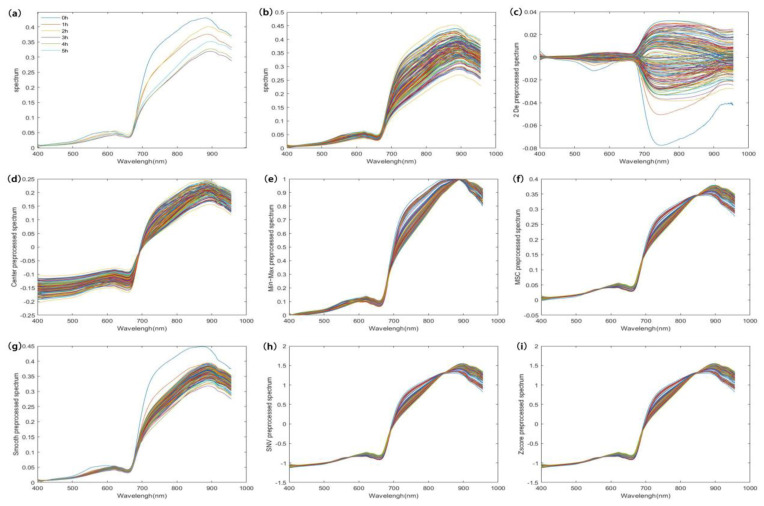
A set of average spectra at different fermentation moments and spectral curves before and after pretreatment. ((**a**) is the average spectrum of black tea at different fermentation times; (**b**) is the original spectrum of black tea at different fermentation moments; (**c**–**i**) are the pre-processed spectra of the original spectra using 2 De, Center, Min-Max, MSC, Smooth, SNV, and Zscore algorithms, respectively).

**Figure 5 sensors-21-08051-f005:**
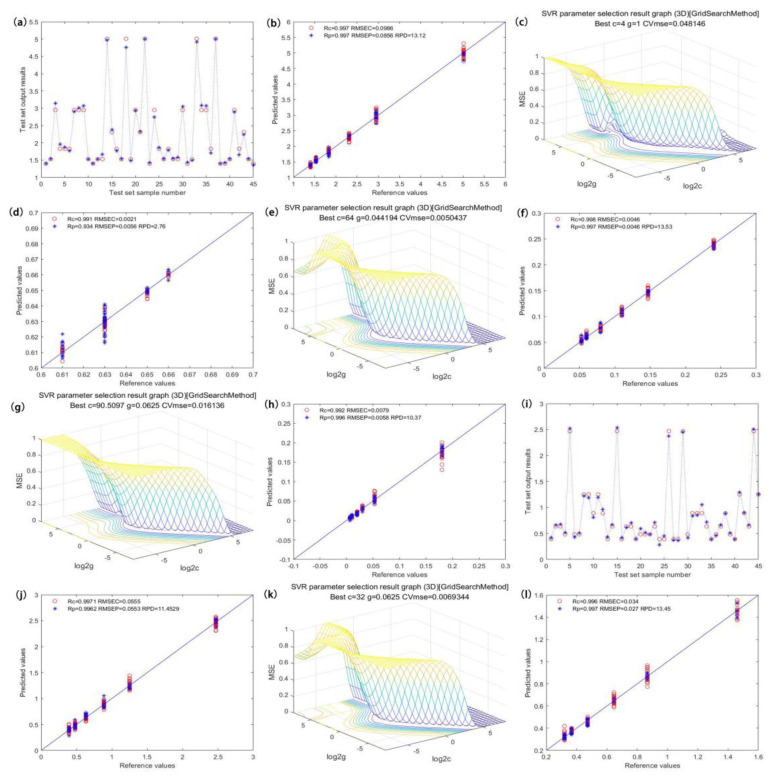
Optimal model optimization results and prediction effects of the content of catechin components. ((**a**,**i**) are the prediction effects of the total catechin model and the EGCG model, respectively; (**c**,**e**,**g**,**k**) are the internal parameter optimization diagrams of EGC, C, EC, and ECG models, respectively; (**b**,**d**,**f**,**h**,**j**,**l**) are the predicted scatter plots of the total catechins, EGC, C, EC and ECG models, respectively).

**Figure 6 sensors-21-08051-f006:**
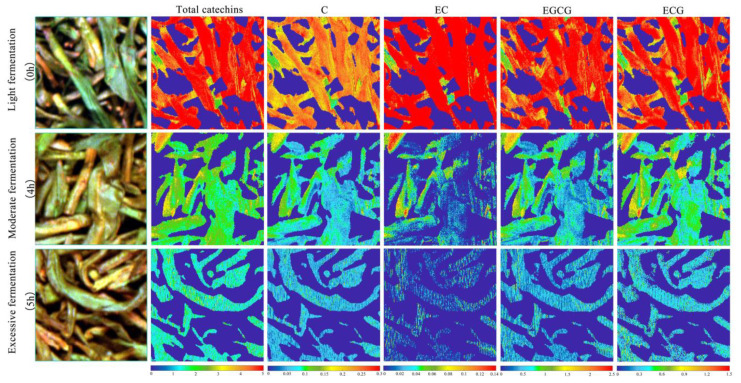
The visual analysis chart of catechin content for different fermentation levels.

**Table 1 sensors-21-08051-t001:** The Characteristic wavelengths screened by different variable screening methods.

Algorithm	Endoplasmic Composition	Characteristic Wavelengths
**SPA**	The total amount of catechins	406, 457, 505, 536, 547, 566, 580, 625, 650, 731, 776, 891, 942, and 947 nm
**VCPA-GA**	EGC	436, 457, 492, 513, 554, 579, 625, 674, 683, 694, 705, 727, 729, 730, 743, 757, 766, 767, 832, 835, 838, 846, 847, 886, 893, 897, 901, 914, 917, 938, 952, and 953 nm
**VCPA-IRIV**	C	417, 418, 435, 441, 442, 486, 507, 526, 548, 556, 614, 615, 623, 679, 696, 707, 786, 789, 797, 841, 887, 904, 905, 927, 928, 941, and 942 nm
**VCPA-IRIV**	EC	436, 441, 447, 497, 499, 557, 558, 607, 627, 638, 656, 667, 683, 689, 696, 786, 883, 886, 887, 901, 904, 955, 956, and 957 nm
**VCPA-IRIV**	EGCG	447, 453, 455, 496, 503, 505, 526, 527, 587, 589, 606, 627, 628, 635, 637, 646, 732, 735, 796, 797, 807, 828, 836, 838, 839, 912, 915, 927, and 948 nm
**VCPA-IRIV**	ECG	447, 453, 455, 496, 503, 505, 526, 527, 587, 589, 606, 627, 628, 635, 637, 646, 732, 735, 796, 797, 807, 828, 836, 838, 839, 912, 915,927, and 948 nm

**Table 2 sensors-21-08051-t002:** Optimal results of pretreatment methods that affect catechin content, in the PLS model.

Physical and Chemical Composition	Pretreatment Method PCs		Calibration Set		Prediction Set	
			Rc	RMSECV	Rp	RMSEP
Total catechins	Z-Score	5	0.918	0.502	0.911	0.592
EGC	MSC	10	0.810	0.0095	0.769	0.0102
C	Z-Score	9	0.889	0.0301	0.883	0.0398
EC	SNV	7	0.903	0.0218	0.891	0.0311
EGCG	Z-Score	9	0.928	0.116	0.920	0.151
ECG	SNV	9	0.929	0.117	0.923	0.140

**Table 3 sensors-21-08051-t003:** Catechin content prediction results from different models.

Catechin Component	Methods	Variable Number	PCs	Calibration Set		Prediction Set		
				Rc	RMSECV	Rp	RMSEP	RPD
Total catechins	SPA-PLS	14	6	0.977	0.268	0.979	0.239	4.46
	SPA-SVR	14	8	0.994	0.142	0.987	0.193	5.92
	SPA-ELM	14	7	0.994	0.136	0.989	0.175	6.50
EGC	VCPA-GA-PLS	32	8	0.946	0.0053	0.956	0.0050	2.03
	VCPA-GA-SVR	32	8	0.983	0.0030	0.972	0.0041	3.78
	VCPA-GA-ELM	32	9	0.954	0.0048	0.926	0.0059	2.66
C	VCPA-IRIV-PLS	27	10	0.993	0.0076	0.991	0.0087	6.11
	VCPA-IRIV-SVR	27	7	0.996	0.0060	0.993	0.0082	6.72
	VCPA-IRIV-ELM	27	9	0.996	0.0056	0.992	0.0086	6.37
EC	VCPA-IRIV-PLS	24	9	0.984	0.0113	0.987	0.0086	4.68
	VCPA-IRIV-SVR	24	7	0.996	0.0059	0.990	0.0075	5.69
	VCPA-IRIV-ELM	24	8	0.995	0.0064	0.988	0.0081	5.25
EGCG	VCPA-IRIV-PLS	29	10	0.991	0.0953	0.991	0.0868	6.24
	VCPA-IRIV-SVR	29	5	0.996	0.0684	0.994	0.0793	7.33
	VCPA-IRIV-ELM	29	7	0.995	0.0701	0.993	0.0825	7.00
ECG	VCPA-GA-PLS	49	9	0.992	0.0496	0.992	0.0498	6.53
	VCPA-GA-SVR	49	8	0.995	0.0426	0.994	0.0502	6.68
	VCPA-GA-ELM	49	7	0.996	0.0335	0.994	0.0468	7.29

## Data Availability

The data sets generated and/or analyzed during the current study are available from the corresponding author on reasonable request.

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
