# Peer review of "Nondestructive Testing and Visualization of Catechin Content in Black Tea Fermentation Using Hyperspectral Imaging"

_sensors, 2021, doi:10.3390/s21238051_

Round 1

Reviewer 1 Report

In this article, authors developed chemometrics models to predict catechin content in fermenting leaves under different fermentation periods using hyperspectral imaging. The findings can provide significant reference for the intelligent and digital processing of black tea. However, there are a few major issues with this manuscript. The author have not clearly reviewed the advantages and disadvantages of hyperspectral imaging. What is the minimal content of catechin in tea that can be determined by hyperspectral imaging. Finally, While not really bad, the English needs to be improved at whole article, especially for methods part.

Specific comments:

There is blank between ENVI and 5.3.

There is blank between model accuracy and (Zhanbin et al. 2020), please check the similar mistakes.

In figure 5-g, why is there only five group for data?

Reviewer 2 Report

Nondestructive Testing and Visualization of Catechin Content in Black Tea Fermentation Using Hyperspectral

In this paper, hyperspectral technology was applied to predict catechin content in black tea fermented under different fermentation periods. Several chemometrics methods and different pretreatments were used to deal with the spectral noise and optimizing the predictions.

The topic itself is interesting and the authors use several chemometric algorithms to optimize the prediction.

Unfortunately, the script is not well written yet.

  1. Title I recommend using Hyperspectral Imaging.
  2. In the abstract, what is the „super normal vector (SNV)”? Do you mean standard normal variate (SNV)?
  3. In the introduction you mention SPA, SPA-MLR, LS-SVR, VCPA-IRIV-RF I suggest inserting into the text what those are.
  4. In Acquisition of experimental samples could you please describe your experiment a little more clearly. „layer-based processing was performed on the stacked leaves”
  5. It is not clear how many samples did you have? The manuscript contains plural but it seem there was one batch. Please clarify it.
  6. In Acquisition of hyperspectral data, what do you mean „resolution as 2.8 nm, the sampling interval as 0.67 nm”? Could you please clarify it?
  7. The conveyor speed was really slow (24 mm/s). Did this cause the sample to heat up? How could you avoid it?
  8. „The average spectra in each ROI were used as the sample spectra at this moment, each
    moment containing 30 spectra, totally 180 spectra.” Please describe your spectral acquisition more clearly.
  9. Figure 1 is too small. I cannot see it.
  10. In Variable screening and dimension reduction „Therefore, this study used seven above-mentioned variable screening algorithms to extract the characteristic wavelengths.”, please list them.
  11. The quality of figure 3 is not acceptable.
  12. In Optimal selection of pretreatment algorithms and in Figure 4 caption, please list what are the pretreatments clearly. Or you maybe do not need figure 4 because it does not give any value.
  13. The quality of Figure 4 is poor.
  14. In Optimal selection of variable screening algorithms what is KS? What is SPA? VCPA-IRIV etc? These needs to be described in briefly preferably in Method section.
  15. „ 5 (a) shows the prediction effect of the total catechin model.” Please clarify it.
  16. I recommend taking apart the Figure 5 to multiple figures.
  17. Figure 6 needs to be explicitly discussed.

Round 2

Reviewer 2 Report

Nice work but the quality of the images still needs to be improved. 
